# Dilated Heterogeneous Convolution for Cell Detection and Segmentation Based on Mask R-CNN

**DOI:** 10.3390/s24082424

**Published:** 2024-04-10

**Authors:** Fengdan Hu, Haigen Hu, Hui Xu, Jinshan Xu, Qi Chen

**Affiliations:** College of Computer Science and Technology, Zhejiang University of Technology, Hangzhou 310014, China; fdhu@zjut.edu.cn (F.H.); hghu@zjut.edu.cn (H.H.); xuhui0681@163.com (H.X.)

**Keywords:** cell detection and segmentation, dilation convolution, heterogeneous convolution, mask R-CNN

## Abstract

Owing to the variable shapes, large size difference, uneven grayscale, and dense distribution among biological cells in an image, it is very difficult to accurately detect and segment cells. Especially, it is a serious challenge for some microscope imaging devices with limited resources owing to a large number of learning parameters and computational burden when using the standard Mask R-CNN. In this work, we propose a mask R-DHCNN for cell detection and segmentation. More specifically, Dilation Heterogeneous Convolution (DHConv) is proposed by designing a novel convolutional kernel structure (i.e., DHConv), which integrates the strengths of the heterogeneous kernel structure and dilated convolution. Then, the traditional homogeneous convolution structure of the standard Mask R-CNN is replaced with the proposed DHConv module to it adapt to shape and size differences encountered in cell detection and segmentation tasks. Finally, a series of comparison and ablation experiments are conducted on various biological cell datasets (such as U373, GoTW1, SIM+, and T24) to verify the effectiveness of the proposed method. The results show that the proposed method can obtain better performance than some state-of-the-art methods in multiple metrics (including AP, Precision, Recall, Dice, and PQ) while maintaining competitive FLOPs and FPS.

## 1. Introduction

Detection and segmentation of cells are two main tasks of automated analysis in the area of biomedical engineering, and they can be of great help to further quantitative analysis of biological cells. For example, quick detection of leucocytes is crucial for early diagnosis of infections. However, owing to the variable shapes and sizes, uneven grayscale, and dense distribution among biological cells in a microscopy image, it is still a challenging task to accurately detect and segment cells. Traditional methods such as thresholding [1,2], edge detection [3], and watershed transform [4,5,6] have been extensively used in automatic cell detection and segmentation tasks. These methods rely on handcrafted features and often require manual tuning of parameters, making them less robust and scalable.

Having witnessed the breakthroughs that deep learning contributed to in various areas, deep learning-based frameworks for detection and segmentation have been continuously proposed and modified for various application scenarios in the biomedical community [7,8,9,10,11]. In fact, worldwide competitions have been established and datasets of cell images under different microscopy conditions have been released to promote deep learning-based technical innovations that enable fast and accurate cell detection and tracking [12]. Among the existing solutions, one-stage object detection algorithms, characterized by the skip of the region proposal module, have high inference speeds with poor performance in accuracy and precision [13,14,15,16]. Two-stage detectors with a region proposal module, for example, Faster R-CNN [17], are very popular owing to their high localization and recognition capabilities [18,19,20,21,22]. By introducing a segmentation branch to Faster R-CNN, the network (called Mask R-CNN) achieved state-of-the-art performance in the task of object recognition [20]. However, good performance in detection and segmentation tasks of Mask R-CNN is achieved at the cost of massive computation, measured as the total amount of Floating Point Operations, FLOPs. The high computational complexity and memory requirement limit its adaptation in remote devices, such as some microscope imaging systems. Consequently, reducing the model complexity while maintaining its high performance is always plausible.

In recent years, research efforts have been devoted to designing more efficient convolution structures that require fewer FLOPs. Heterogeneous convolution (HetConv) was one of the attempts that aimed at reducing computations and model parameters while achieving faster processing capability (measured in terms of Frames Per Second, FPS) and maintaining the same representational efficiency as the standard convolution operations. However, the HetConv structure contains convolution kernels of a size of 1×1, which could easily lead to a lack of local information capture capability and result in poor detection and segmentation performance. Consequently, compensating for the loss of local information caused by HetConv with more spatial information is demanding for cell detection and segmentation tasks.

Regarding the importance of a large perceptive field in object detection tasks and the requirement of a light network model, the dilated convolution operation was invented to enlarge the receptive field by regularly inserting zeros between the kernel elements [23]. This dilated kernel enables convolution operations over values spanning larger distances without introducing extra parameters. Apparently, the combination of the advantages of the above network structures would be an ideal solution for detection and segmentation tasks.

To this end, we propose a novel convolutional kernel structure called Dilation Heterogeneous Convolution (DHConv), which integrates the advantages of the heterogeneous kernel structure and dilated convolution. By replacing the traditional convolutional kernel of the standard Mask R-CNN with the proposed DHConv module, we then propose a novel deep neural network called Mask R-DHCNN for detection and segmentation of objects of variable shapes and sizes, such as cells in different microscopy images. We demonstrate the effectiveness of the proposed solution by carrying out tests on the publicly available and most frequently used microscopy cell image datasets. Our main contributions are as follows:We propose a novel convolutional kernel structure called DHConv that combines the advantages of dilated convolution and HetConv.An improved Mask R-CNN, called Mask R-DHCNN, is proposed by replacing the standard convolution of the original Mask R-CNN with the proposed DHConv module, which leads to easy adaptation of variable shapes and sizes in the task of cell detection and segmentation.A series of experiments are conducted to verify the effectiveness of the proposed methods on various datasets, and the results show that the proposed method can obtain better performance than some state-of-the-art methods in multiple metrics while maintaining competitive FLOPs and FPS.

The rest of this paper is organized as follows: Section 2 introduces the work related to the current study. Section 3 details the proposed methods, including DHConv and Mask R-DHCNN. Section 4 presents the experimental settings and the obtained results, and Section 5 concludes this work.

## 2. Related Work

### 2.1. Dilated Convolution

Dilated convolution, also called Atrous Convolution or convolution with holes, was initially introduced for semantic segmentation [24,25]. This convolution operation has been widely used in object detection and semantic segmentation [26]. The most significant advantage of dilated convolution is the enlarged receptive field brought by expansion of the receptive size without any increase in Float Point Operations or additional loss of resolution. Compared to pooling operations with similar receptive field expansion, dilated convolution can significantly improve performance for the detection and segmentation of multi-scale objects [27].

To quantitate the enlargement of the receptive field brought by dilated convolution, the dilation rate *r* is typically introduced to adjust the expansion range of the receptive field. This dilation rate *r* defines the spacing between the values in a kernel. Standard convolution (with no additional space between kernel values) is a special case of dilated convolution with dilation rate r=1, and dilating the normal convolution kernel of size $K_{N}$ with rate *r* could result in a dilated convolution kernel of size $K_{D}$ as
(1)$$K_{D}=\left(K_{N}-1\right)\cdot r+1$$
That is to say, easy regulation of the sampling range of the convolution kernel can be achieved through the dilation ratio *r*.

### 2.2. HetConv

HetConv (Heterogeneous Kernel-Based Convolution) [28] was proposed to reduce the required computations (FLOPs) and the number of model parameters. This structure obtained competitive performance while maintaining the same representational efficiency compared with the standard convolution operation. Traditional convolutions (including standard convolution, group convolution, and depth-wise convolution) usually use a homogeneous filter where each kernel is of the same size throughout the whole filter. HetConv adopts a Heterogeneous Convolution structure where the composed kernels are of different sizes. More precisely, the first kernel keeps the size of K×K and the remaining kernels are replaced by small kernels of size 1×1. We introduce the parameter *P* to describe the ratio between the number of kernels of size K×K and total number of kernels in a filter. Recall that the original kernels are all of size K×K; HetConv replaces a fraction of 1−P of the original K×K kernels with 1×1 kernels, leading to an efficient reduction in the amount of calculations.

Suppose that the input feature map of $D_{w}\times D_{h}\times N$ dimensions is convolved with *N* standard K×K×C filters; the computational cost on the layer *L* is provided as
(2)$$FL_{c}=D_{w}\times D_{h}\times N\times K\times K\times C$$
When some (1−P) of the original K×K convolution kernels are replaced by the 1×1 kernels, the cost of K×K convolution kernels is provided as
(3)$$FL_{k}=\left(D_{w}\times D_{h}\times N\times K\times K\times C\right)\cdot P$$
and the cost of 1×1 convolution kernels is provided as
(4)$$FL_{1}=\left(D_{w}\times D_{h}\times C\right)\left(N-NP\right)$$
Then, the total cost of the HetConv is calculated as
(5)$$FL_{hc}=FL_{k}+FL_{1}$$
Comparing Equations (Equation 2) and (Equation 5), we can obtain the ratio $R_{hc}$ of the computational cost between HetConv and standard convolution as follows:(6)$$R_{hc}=\frac{FL_{hc}}{FL_{c}}=P+\frac{\left(1-P\right)}{K^{2}}$$

When P=1, the HetConv is equivalent to the standard convolution. When P<1, the computational cost of the HetConv is determined by combined effect of parameter *P* and kernel size *K*.

The analysis above implies that the Heterogeneous Convolution can reduce the computational cost by introducing convolution kernels of different sizes to the convolution filters. However, without losing the accuracy of the model, the HetConv actually contains plenty of 1×1 convolution kernels, which impairs its ability in extracting local information. As a result, the advantages of a heterogeneous structure could not be fully utilized in HetConv [28].

### 2.3. Mask R-CNN

As an improved version of the classical Faster R-CNN [17], Mask R-CNN is a more efficient multi-task model [20]. To enable mutli-scale feature extraction of an object, Mask R-CNN introduces a feature pyramid network (FPN) [19] on top of the conventional feature extraction network. This top-down structure and horizontal connection help in fusing the feature maps in the conventional neural network. In addition, The Region of Interest (RoI) pooling operation of Faster R-CNN is replaced by RoIAlign [29], which uses a bilinear interpolation method to align the object in a pixel-by-pixel way [30], leading to improved segmentation accuracy. Lastly, the proposed regions are sent directly to the full connection network (FCN) to achieve semantic segmentation by introducing segmentation branches so that a unique segmentation mask is generated for each object. This combination of detection and segmentation improves the model performance of both detection and segmentation tasks. In addition, the integrated loss of different tasks, i.e., classification, regression, and segmentation, to represent the overall loss of the network model also improves model performance.

In fact, Mask R-CNN increases performance accuracy at a cost of increased model complexity brought by a two-stage framework and a multi-task branch structure, resulting in a tremendous increase in computation (FLOPs). Therefore, how to reduce computational costs to make the model more efficient is still a challenging task. In this work, we attempt to overcome the limitation of the existing Mask R-CNN for the task of cells detection and segmentation by integrating dilated convolution into the heterogeneous structure based on Mask R-CNN.

## 3. Proposed Method

As mentioned above, HetConv provides competitive performance while still maintaining representational efficiency owing to the heterogeneous structure of convolution kernels in a filter. However, the existence of too many 1×1 convolution kernels in each filter also provides adverse effects, i.e., loss of valuable information when extracting features on the adjacent regions of an image in the object detection and segmentation tasks. Specifically, this shortcoming poses constraints to its application in cell detection and segmentation tasks, where the objects (cells) are of a variety of shapes and large size difference. Normally, introducing large convolution kernels could resolve the adverse effect brought by small receptive field of 1×1 kernels. However, large convolution kernels would inevitably provide increased computing burden. Considering the limited computing capability of some remote devices, it is important to design network solutions that are light and highly efficient. To this end, we combined the advantages of HetConv and dilated convolution and implemented them in the Mask R-CNN network.

### 3.1. Dilation Heterogeneous Convolution (DHConv)

DHConv is proposed by integrating the advantages of the dilated convolution and HetConv. This new network structure borrows the idea of the efficient heterogeneous architecture containing different size kernels for the same filter and the dilated convolution to overcome the limitation of insufficient feature extraction capability caused by 1×1 convolution in HetConv. Owing to the larger receptive field brought by the dilated kernel, reduction in information caused by the heterogeneous convolution could be avoided. In DHConv, the K×K convolution kernels of the original HetConv are replaced by the dilated convolution kernels of dilation rate *r*. This provides expansion to the receptive field to size o [(K−1)×r+1]×[(K−1)×r+1]. Figure 1 illustrates the spatial layout of the dilated heterogeneous kernels, where the original *M* convolution kernels are replaced by M·P dilated convolution kernels of dilation rate *r* and (1−P)×M convolution kernels with a size of 1×1.

A detailed comparison between HetConv and the proposed DHConv is illustrated in Figure 2. By introducing *r* holes to the original convolution kernel, DHConv gains a significant expansion in its receptive field. Compared with the original HetConv, the proposed DHConv uses the dilated convolution kernels in a heterogeneous spatial manner. Apparently, this layout does not add any additional parameters and requires no additional computational resources, while it can provide larger coverage when extracting features from images. Generally, the more 1×1 convolution kernels in a filter, the lower parameters and computational burden in the model. DHConv inherits the advantages of 1×1 convolution operations by leveraging heterogeneous kernels. Therefore, the proposed method is computationally efficient. According to the analysis in Section 2.2, the proposed DHConv could reduce the computational cost by a ratio of $R=P+\left(1-P\right)/K^{2}$ and enlarge the size of the receptive field by a factor $∼r^{2}$, with *P* and *r* being the ratio of number of large kernels in the HetConv and dilation rate, respectively. In addition to the benefits from the structural changes, the proposed DHConv has less memory consumption as it can skip the pooling step.

### 3.2. Mask R-DHCNN

Mask R-CNN is currently the most influential instance segmentation framework. As mentioned above, Mask R-CNN can achieve state-of-the-art detection and segmentation performance at the cost of a tremendous increase in computation (FLOPs). Applying this framework directly to devices with limited resources to detect and segment cells might not provide benefits. To address this issue, it is quite demanding to improve Mask R-CNN with the purpose to increase its computing efficiency and maintain competitive FLOPs and FPS. Considering the fact that feature extraction (i.e., ResNet backbone, composed of a set of convolutional filters) is the aspect with the highest computing resources consumption, we could reduce the computing burden by replacing the normal homogeneous structure with the heterogeneous structure in the ResNet backbone. To this end, an improved Mask R-CNN, called Mask R-DHCNN, is proposed by integrating the proposed DHConv into the original Mask R-CNN in this work. Specifically, the normal homogeneous convolutions of the ResNet in the original Mask R-CNN are replaced with dilated Heterogeneous Convolutions (i.e., DHConv). Our analysis presented previously implies that the proposed framework could effectively overcome the limitation of the original architecture that requires large number of learning parameters and computational burden.

Compared with the original Mask R-CNN, the proposed Mask R-DHCNN can greatly enlarge the receptive field and can reduce FLOPs regarding the model more efficient owing to the heterogeneous structure, thereby providing benefits as follows: (i) Mask R-DHCNN uses larger area of image in extracting features for cell recognition without a large extraction computation cost owing to the heterogeneous spatial layout. (ii) An expansion in the receptive field is achieved by introducing dilated convolution, which effectively compensates for the loss in local information caused by 1×1 convolution under heterogeneous kernel structures.

## 4. Experiments and Results

### 4.1. Experimental Settings

The proposed Mask R-DHCNN uses ResNet-50-FPN [31] as its backbone and the the corresponding Mask R-CNN as the baseline. We first implemented Heterogeneous Convolution by replacing the last three of the four 3×3 kernels in the filters of the backbone structure by 1×1 kernels; i.e., P=1/4 and 3/4 of conventional kernels of size K=3 were replaced by the small kernels of size 1 × 1. Recall the theoretical estimation provided in Equation (Equation 6): the computational cost for implementing the DHCovn module is only 1/3 of the original convolution operations. So, a reduction in computational costs as large as 2/3 could be expected. In addition, to compensate for the adverse effect from small kernels as well as too-large kernels, dilation with rate r=2 was applied to the 3×3 kernels.

We implemented the proposed Mask R-DHCNN under the Pytorch framework and tested its performance in cell detection and segmentation tasks on a Windows 10 OS-based laptop equipped with an Inter(R) Core(TM) i7-9750H CPU (2.6 GHz) 6-core 12-thread processor and a GeForce GTX 2060 6G graphics card. GPU implementation of the SGD (Stochastic Gradient Descent) optimizer was used to accelerate the forward propagation and back propagation routines. The same parameters were used throughout all the experiments shown below. Meanwhile, a variable learning rate was adopted, with an initial value of 0.001 and the rate of change 0.1 every 10 epochs. The weight decay rate and maximum number of iterations were set to 0.0005 and 50 epochs, respectively. The average over three executions was used to evaluate the model performance.

### 4.2. Datasets

Cell tracking, a time-consuming and tedious task, has always been of great interest to biologists. Developing computer programs that could perform automatic cell tracking is quite demanding. To promote such a technical breakthrough, international challenges on Cell Tracking has been established since 2014, alongside the release of datasets of cell images under different microscopy environments [12]. Among them, PhC-C2DH-U373 (short for U373), Fluo-N2DH-GOWT1 (short for GOWT1), and Fluo-N2DH-SIM+ are frequently used in cell detection and segmentation tasks [11,14,22,32,33] as they are the pre-steps for performance tracking tasks. To set a baseline for comparison, we adopted these datasets to demonstrate the effectiveness of the proposed Mask R-DHCNN framework. These three datasets provide cell images obtained from a set of typical microscopy conditions. For example, the cells in U373 are deformed, some of which do not even have round shapes. Images in GOWT1 typically contain more cells. Particularly, the Fluo-N2DH-SIM+ dataset contains simulated cells with (Fluo-N2DH-SIM+-01, short for SIM+-01) and without fluorescent chromosomes (Fluo-N2DH-SIM+-02, short for SIM+-287 02). Moreover, we also include human bladder cancer cells T24 images from a series of phase contrast microscopy videos from the Cancer Cell Institute, University of Cambridge. Figure 3 shows a set of representative microscopy cell images in these datasets. Detailed descriptions of the datasets are provided below.

PhC-C2DH-U373 (U373) comprises images of glioblastoma–astrocytoma cells obtained through phase contrast microscopy. These cells are cultured adherently and video recorded under controlled conditions to provide data for cell tracking and analysis tasks. Due to only subtle differences between two consecutive frames, not all the frames can be considered as training samples for the proposed network. To address this issue, we select one frame from two adjacent images (video frames) as a sample. This operation provided us 230 sample images, among which 184 and 46 images are, respectively, used for training and testing.Fluo-N2DH-GOWT1 (GOWT1) was obtained from GFP transfected GOWT1 mouse embryonic stem cells on a flat substrate using a Leica TCS SP5 laser scanning confocal microscope (courtesy of Dr. E. Bártová, Academy of Sciences of the Czech Republic, Brno, Czech Republic). This dataset exhibits characteristics of low contrast and high density under microscope due to factors such as heterogeneous staining, prominent nucleoli, mitoses, cells entering and leaving the field of view, and frequent cell collisions. The dataset also has two image sequences with a total of 368 frames. Similarly, we extract one frame from every two frames and can obtain 184 frames, 147 images for training and 37 images for testing.Fluo-N2DH-SIM+ contains simulated cells stained with and without fluorescent chromosomes. The morphology of the cells between the two conditions is quite different. Therefore, both the stained and non-stained images are taken as two separate datasets, Fluo-N2DH-SIM+-01 (short for SIM+-01) and Fluo-N2DH-SIM+-02 (short for SIM+-02). In Fluo-N2DH-SIM+-01, cells are of more complex morphology and low fore-back ground contrast exists. In Fluo-N2DH-SIM+-02, the scale of cells varies greatly, and there are partial occlusions between the cells, which greatly increases the difficulty of segmentation. Moreover, 52 frames and 13 frames from Fluo-N2DH-SIM+-01 (SIM+-01) were selected as training set and testing set, respectively. In Fluo-N2DH-SIM+-02, 120 frames and 13 frames were used for training and testing, respectively.T24 is a real bladder cancer cell dataset released by the Cancer Cell Institute, University of Cambridge. The cells were cultured in RPMI-1640 medium (HyClone, Logan, UT, USA) supplemented with 10% heat-inactivated FBS (JRH Biosciences, Lenexa, KS, USA), 100 U/mL penicillin, and 100 mg/L streptomycin. Cultures were maintained in a humidified atmosphere of 5% CO_2_ at 37 °C. Adhesion and overlap between cells were common in these images. In addition, the size of cells varies greatly, making detection and segmentation challenging.

### 4.3. Data Annotation

Recalling that the microscopy images in these publicly available datasets released from the ISBI competition have not been labeled, we adopted MS COCO2014, released by the Microsoft team [34], for sample labels. It provided us with a dataset format that is suitable for detection and segmentation tasks. The label consists mainly of five parts: (1) the info field is mainly used to store the descriptions on the dataset, including data provider, and the year when the data were marked; (2) the licenses field stores the license information of the dataset; (3) the image field contains information such as ID, name, size, and generation time of the image; (4) the annotations field stores a list of multiple annotation instances, which can, respectively, show the category ID and segmentation mask of each object, so the ground truth required for the experiment can easily be extracted; (5) the categories field provides conclusion information on each category in the image. We took Labelme [35] as the data annotation tool, which enables us to mark out the specific contour position of the object in the original image and generate the corresponding JSON file with object location information. Figure 4 illustrates the effect of annotation for T24 dataset.

### 4.4. Evaluation Metrics

Recalling the purpose of developing automatic cell detection and segmentation solutions suitable for remote device implementation, we chose Average Precision (AP), Precision, Recall, Dice, Panoptic Quality (PQ), FLOPs, and FPS as the evaluation metrics to quantitatively demonstrate the effectiveness of the proposed Mask R-DHCNN. Among them, AP, Precision, and Recall are frequently used evaluation metrics in image classification and object detection tasks, while Dice is specific for segmentation tasks. PQ was adopted in this work to demonstrate the model’s comprehensive performance on both detection and segmentation tasks. Apparently, the first four metrics quantify the performance of Mask R-DHCNN in cell detection and segmentation tasks, while FLOPs and FPS were used to demonstrate execution efficiency. Typically, the more Floating Point Operations required (high FLOPs values), the fewer frames the Mask R-DHCNN can process within a second (low FPS). For illustrative purposes, we provide the definitions of the metrics in the following. By introducing the IoU (Intersection over Union), defined as the ratio between the overlapped area between model prediction and ground truth and the union of the two, we can determine the basic metrics with the threshold IoU as follows

TP: real cells that are correctly identified as real cell objects;TN: non-cell objects that are correctly identified as non-cell objects;FP: non-cell objects that are mistakenly identified as cell objects;FN: real cells that are mistakenly identified as non-cell objects.

With the basic quantities defined, the evaluation metrics are calculated as

Precision: a measure of the percentage of the correctly classified cell objects in all classifications of cell objects, which is the probability of not misclassifying cell object, i.e.,
(7)$$Precision=\frac{TP}{TP+FP}$$Recall: a measure of the percentage of the correctly classified cell objects in the total real cells, which is the probability of non-missed diagnosis, i.e.,
(8)$$Recall=\frac{TP}{TP+FN}$$Dice: a measure of the proportion of intersection between the model output and the ground truth, i.e.,
(9)$$Dice=\frac{2\times TP}{2\times TP+FN+FP}$$AP: Recall and Precision are contradictory in evaluating object detection performance as both of them depend on the threshold value of IoU we take to set the TP, FN, TN, and FP. To this end, we calculate the area under the Precision–Recall curve constructed by applying different threshold values to IoU as the average precision of the model, i.e.,
(10)$$AP=\int_{0}^{1}P\left(\gamma \right)d\gamma$$
where γ and P(γ) are the Recall and the dependent Precision, respectively.PQ: is commonly used in medical image segmentation tasks. Although Dice is a good index of the model’s segmentation ability, it cannot reflect model performance in both detection and segmentation tasks. Simon et al. [36] proposed PQ that is capable of illustrating the pros and cons of the model in performing detection and segmentation tasks. The definition of PQ is as follows:
(11)$$PQ=\left\{\begin{array}{c} \underset{︸}{\frac{TP}{TP+\frac{1}{2}FN+\frac{1}{2}FP}}\\ Detection\;Quality(DQ) \end{array}\right\}\times \left\{\begin{array}{c} \underset{︸}{\frac{\sum_{(x,y)\in TP}IoU\left(x,y\right)}{TP}}\\ Segmentation\;Quality(SQ) \end{array}\right\}$$
where *x* and *y* are the model prediction and the ground truth, respectively.

Different from the Average Precision (AP), Precision, Recall, Dice, and PQ, which have to be calculated by comparing to the ground truth, the FPS and FLOPs are provided by Pytorch directly.

### 4.5. Experiments and Results

The potential advantages obtained from the theoretical analysis of the proposed model motivated a set of experiments that serve as solid evidence of the superiority of the proposed Mask R-DHCNN structure. In Figure 5, we presented typical detection and segmentation results when applying the proposed method to images in datasets T24,U373, GoTW1, SIM+01, and SIM+02. Alongside the results, the ground truth images are displayed on the left for direct visual comparison. One can see from this figure that the proposed model could label the existence of cells in microscopy images taken from different environmental sets.

A visual experiment on a microscopy image that contains cells of different shapes and sizes has verified the effectiveness and robustness of the proposed method in segmentation tasks. As illustrated in Figure 6, the model’s prediction of the cell body almost lies exactly on the manually labeled region, where the yellow region marks the overlap between the model prediction and manual label, the green region marks the labeled cell body that has not been predicted, and the red region is the predicted cell region that does not have any cell.

#### 4.5.1. Comparison Experiments

To verify the effectiveness of the proposed method, a series of comparison experiments were conducted on various cell datasets by adopting different instance segmentation methods, such as Mask R-CNN [20] (baseline), MS R-CNN [22], ExtremeNet [32], TensorMask [14], and the two most recent ones, PolarMask [33] and ResNet-50-FPN-ISO [11]. Among these comparison methods, Mask R-CNN [20] and MS R-CNN [22] are both anchor-based two-stage instance segmentation algorithms. More specifically, MS R-CNN is an improved version of Mask R-CNN by adding a single MaskIoU for Mask score, which can obtain similar performance to the original Mask R-CNN. Meanwhile, ExtremeNet [32] and TensorMask [14] both belong to anchor-free instance segmentation methods.

Table 1 presents a detailed comparison regarding the performance of the proposed method with respect to the representative state-of-the-art solutions. One can see from the table that the proposed method provides quite competitive performance in almost all metrics with respect to the recent state-of-the-art methods. As one of the best instance segmentation methods, the two-stage structure-based Mask R-CNN (our baseline) is superior to the other methods in the detection tasks, whose performances are typically measured using metrics such as AP, Precision, Recall, and Dice. The proposed method outperforms the baseline (Mask R-CNN) by 0.11–2.07 on various datasets. Compared with other SOTA methods, the performance of the proposed method is also very competitive. Although the MS R-CNN achieved the best performance scores in terms of AP, Precision, Recall, and Dice regarding dataset SIM+02, its application to other datasets was not very appealing. The most possible reason is that the cells in the SIM+02 dataset are very suitable for the mask scoring strategy of the MS R-CNN to extract features effectively. As expected, in terms of the integrated use of the advantages of the heterogeneous kernel structure that reduces the computational burden, the proposed Mask R-DHCNN showed faster computing speed, indicated by a higher FPS value. Our results show consistency with previous work (HetConv) [28] that demonstrated heterogeneous kernel structures can improve the computational efficiency and number of parameters as compared to standard convolution operations while still maintaining representational efficiency.

#### 4.5.2. Ablation Experiments

To further verify the effectiveness of the proposed DHConv module, a series of ablation experiments were conducted by taking different modules out and testing on various datasets. The obtained results are illustrated in Table 2. Compared with Mask R-CNN+HetConv, the proposed Mask R-DHCNN (i.e., Mask R-CNN+DHConv) provided the highest PQ value in these test datasets, demonstrating its strong capability in performing unified detecting and segmenting tasks. Although the introduction of HetConv did not reduce the computational cost as much as estimated in Equation (Equation 6), as the estimation was provided by taking solely the convolution operations into account, the benefits from HetConv were confirmed by a much smaller FLOPs value in comparison to that of the baseline. The small kernel convolutions in HetConv brought negative effects to the ability of feature extraction due to a limited respective field (see the PQ columns). Thanks to the dilated convolution we introduced to the original HetConv, extra gains in the abilities of detection (significant increase in PQ values) and segmentation were witnessed without extra expenditure in computational costs or slowdown.

As mentioned above, HetConv contains a large number of 1×1 convolutions and only a small number of 3×3 convolutions, leading to insufficient capabilities in local relationship modeling. The dilation convolution (DConv) simply expands its kernel size by introducing holes to the kernel, resulting in so-called “gridding artifacts” [37]. Such operation significantly reduces its ability to distinguish cells from their environment. As no additional module was introduced, the Baseline+ DConv did not require additional FLOPs. However, as the dilated kernels have to search for pixels located in larger areas, a small reduction in FPS was observed. In the proposed DHConv, although there were as many 1×1 convolutions as in HetConv, the original receptive field of 3×3 kernel convolutions was expanded by the dilated convolution. This replacement compensated effectively for the loss in local information caused by 1×1 convolution in the heterogeneous kernel structures. Therefore, our method only expanded the receptive field size without any increase in computational cost or accruing loss in terms of capability in detection and segmentation tasks. Notably, the reduction in computational costs (FLOPs) obtained from the introduction of the HDConv module was less than the theoretical expectation. This inconsistency resulted from the ignorance of other components in the backbone Mask R-CNN structure when performing the analysis of Equation (Equation 6). Consequently, when the backbone becomes larger, the benefits of HDConv could be less obvious.

## 5. Conclusions and Discussion

Detection and segmentation of cells can be of great help regarding further quantitative analysis of biological cells. However, it is still a challenging task to accurately detect and segment cells owing to the variable shapes and sizes, uneven grayscale, and dense distribution among biological cells in an image. Although deep learning-based techniques have achieved great breakthroughs in the area of computer vision, developing solutions suitable for performing automatic cell detection and segmentation tasks is still challenging due to the large number of learning parameters and computational burden. To address these issues, a novel convolutional kernel structure (i.e., DHConv) was first proposed by integrating the advantages of the heterogeneous kernel structure and dilated convolution in this paper. Then, Mask R-DHCNN was proposed by replacing the traditional convolutional kernel in the standard Mask R-CNN with the proposed DHConv module to make it adapt to variable shapes and large size differences in cell detection and segmentation tasks. A series of experimental results were conducted to verify the effectiveness of the proposed method. The obtained results verified that the proposed Mask R-DHCNN can obtain better performance than some state-of-the-art methods in AP, Precision, Recall, Dice, and PQ while maintaining competitive FLOPs and FPS. It is promising and encouraging for real-world applications of biomedical engineering in the future.

Notably, the images we used to validate the effectiveness of the proposed method were extracted from standard databases; the excellent performance demonstrated in the paper might not guarantee the same performance in all real-world applications. Since the images in the database were typically collected under some specific conditions, the network trained with these images might need finetuning to adapt to real-world scenarios. A common case is the difference caused by different scales as some detailed morphological structures are not visible under small scales. Unfortunately, there is no scale information of the images provided in the database. As a result, no suggestions can be provided to biologists on how to collect microscopy images. Another case that biologists would encounter is dust contamination. In the current study, although we have not explicitly discussed how the proposed method differentiates dust from cells as the performance metrics were provided by direct comparison to the ground truth, the results herein might imply that acceptable performance would be available in real applications.

Mask R-CNN may struggle to capture fine-grained local information. In the application of object detection tasks, anchor boxes that capture the scale and aspect ratio of specific objects are typically used to reduce the cost of the sliding window approach. The proposed method might be less effective in detecting and segmenting cells of irregular morphological structure compared to those of common regular shapes. Using the predefining anchor boxes to catch up all the irregularity of the cell morphology is challenging. Clustering methods that generate anchor boxes with different aspect ratios through adaptive learning would be incorporated in the network to handle those cells of irregular shapes [11]. Our work can serve as a new tool for biologists in their studies on biological or clinical purposes, and inspire novel technical innovations that are capable of other tasks, such as distinguishing different cell types. To this end, we made our codes available on Github: https://github.com/HuHaigen/Mask-R-DHCNN. We hope that a more efficient model can be built based on our work as we are sure that there are many difficulties to integrate the model in real microscopy settings directly. 

## Figures and Tables

**Figure 1 sensors-24-02424-f001:**
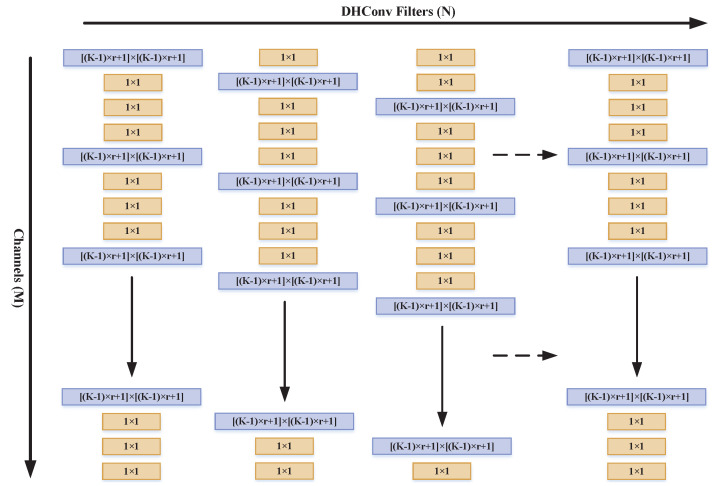
Structure of the dilated Heterogeneous Convolutional Kernels (DHConv). The horizontal axis and vertical axis, respectively, represent *N* DHConv filters and *M* channels in a DHConv filter, in which the number of (1−P)×M convolution kernels in each filter is replaced with 1×1 kernels and the remaining convolution kernels use M·P dilated convolution with a dilation rate *r*.

**Figure 2 sensors-24-02424-f002:**
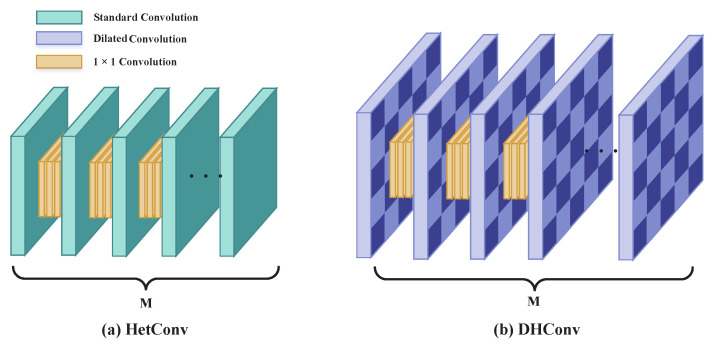
The spatial layout comparison of the convolutional kernels in a filter between HetConv and the proposed DHConv.

**Figure 3 sensors-24-02424-f003:**
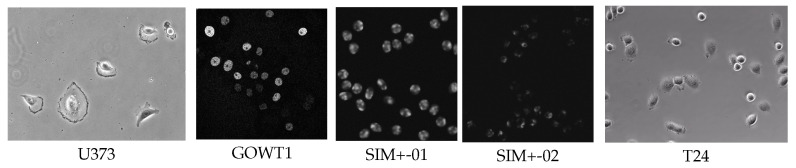
Illustrations of different cell datasets.

**Figure 4 sensors-24-02424-f004:**
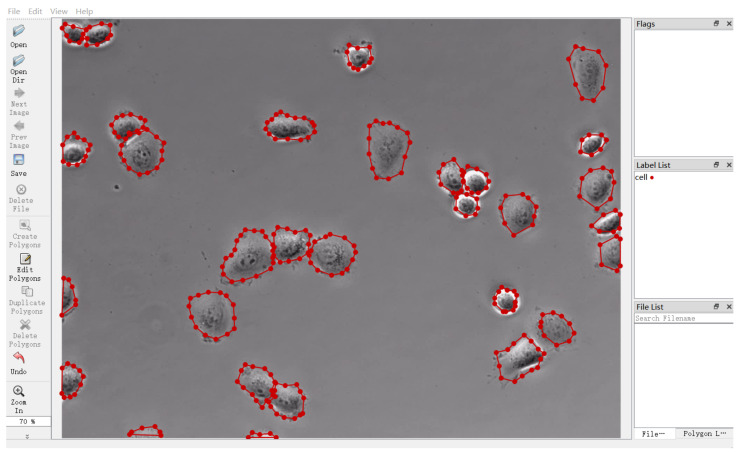
An example illustration of T24 dataset by using Labelme for sample annotation.

**Figure 5 sensors-24-02424-f005:**
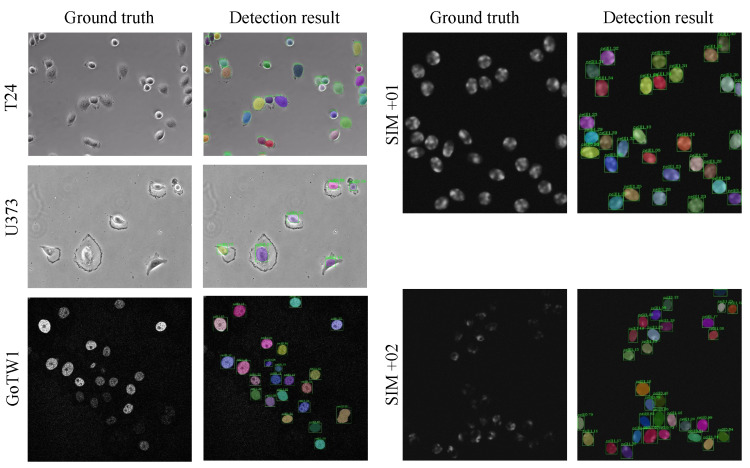
Typical detection and segmentation results of the proposed method applied on different datasets.

**Figure 6 sensors-24-02424-f006:**
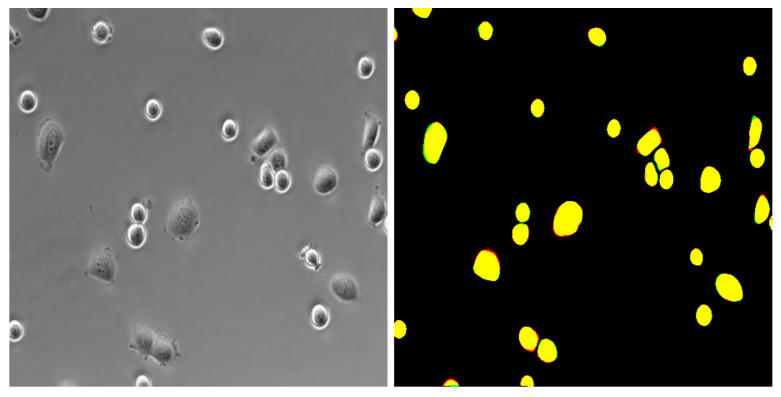
Model’s segmentation performance regarding cells with different morphologies.

**Table 1 sensors-24-02424-t001:** Comparisons of cell detection and segmentation on various datasets between different methods, including baseline (Mask R-CNN [20]), MS R-CNN [22], ExtremeNet [32], TensorMask [14], PolarMask [33], CenterMask [13], ResNet-50-FPN-ISO [11], and our method. The top results are highlighted in bold.

Datasets	Methods	AP (%)	Precision (%)	Recall (%)	Dice (%)	FPS
U373	Baseline	91.39 ± 0.33	87.21 ± 0.30	79.31 ± 0.27	82.03 ± 0.47	4.81
	MS R-CNN	90.12 ± 0.12	86.35 ± 0.17	78.90 ± 0.25	81.87 ± 0.31	4.81
	ExtremeNet	77.75 ± 1.11	71.68 ± 1.27	60.55 ± 0.89	70.11 ± 0.77	4.73
	TensorMask	83.37 ± 1.51	79.92 ± 1.93	68.31 ± 2.11	78.41 ± 0.88	2.47
	PolarMask	88.77 ± 0.10	83.09 ± 0.07	71.93 ± 0.21	80.85 ± 0.13	**11.79**
	CenterMask	79.33 ± 1.74	72.40 ± 1.82	61.40 ± 1.19	74.90 ± 1.85	7.15
	ResNet-50-FPN-ISO	92.74 ± 0.47	**88.65 ± 0.19**	**83.04 ± 0.27**	82.81 ± 0.31	-
	Mask R-DHCNN (**Ours**)	**92.87 ± 0.53**	88.26 ± 0.06	80.52 ± 0.18	**84.21 ± 0.80**	7.00
GoTW1	Baseline	90.64 ± 0.44	91.14 ± 0.48	87.66 ± 0.23	89.65 ± 0.33	4.00
	MS R-CNN	88.77 ± 0.64	89.26 ± 0.88	85.38 ± 0.72	86.05 ± 0.59	3.95
	ExtremeNet	84.40 ± 1.03	86.75 ± 1.21	80.51 ± 0.89	82.37 ± 1.22	3.90
	TensorMask	80.09 ± 0.97	76.27 ± 1.17	70.44 ± 0.82	76.27 ± 1.19	2.00
	PolarMask	85.65 ± 0.83	87.00 ± 0.77	83.43 ± 0.50	85.98 ± 0.81	**9.50**
	CenterMask	78.10 ± 1.56	74.51 ± 2.10	67.39 ± 1.68	73.71 ± 1.40	6.13
	ResNet-50-FPN-ISO	91.18 ± 1.07	**92.26 ± 0.89**	**90.99 ± 1.14**	**91.05 ± 0.59**	-
	Mask R-DHCNN (**Ours**)	**91.26 ± 0.85**	91.84 ± 1.23	88.99 ± 0.94	90.61 ± 0.65	6.70
SIM+01	Baseline	93.93 ± 0.69	94.06 ± 0.21	86.18 ± 0.58	87.60 ± 0.40	4.10
	MS R-CNN	92.03 ± 0.27	93.10 ± 0.89	85.86 ± 0.33	86.38 ± 0.64	4.00
	ExtremeNet	88.64 ± 1.45	90.49 ± 1.29	81.30 ± 1.37	83.24 ± 1.01	3.84
	TensorMask	87.24 ± 0.93	89.94 ± 1.39	80.80 ± 1.27	83.05 ± 1.71	2.15
	PolarMask	91.19 ± 1.13	92.08 ± 0.97	84.65 ± 0.48	85.74 ± 0.70	**10.05**
	CenterMask	85.31 ± 1.66	88.38 ± 1.02	80.29 ± 1.85	78.77 ± 1.90	6.20
	ResNet-50-FPN-ISO	**94.87 ± 0.44**	**94.79 ± 0.39**	84.67 ± 0.61	89.66 ± 0.57	-
	Mask R-DHCNN (**Ours**)	94.04 ± 1.23	94.36 ± 0.87	**88.03 ± 0.42**	**90.13 ± 0.54**	5.50
SIM+02	Baseline	80.88 ± 1.05	83.95 ± 1.06	80.69 ± 1.88	75.71 ± 1.24	3.81
	MS R-CNN	**88.43 ± 1.07**	**87.92 ± 1.21**	**85.49 ± 1.53**	**83.10 ± 1.56**	3.75
	ExtremeNet	73.22 ± 2.71	72.49 ± 1.88	70.20 ± 2.47	70.17 ± 1.06	3.50
	TensorMask	75.41 ± 0.91	74.18 ± 0.54	70.77 ± 1.23	71.20 ± 1.51	2.30
	PolarMask	78.52 ± 0.99	79.06 ± 1.15	74.36 ± 1.22	74.18 ± 0.85	**9.23**
	CenterMask	70.63 ± 2.92	69.30 ± 3.05	67.27 ± 1.87	66.98 ± 1.28	5.75
	ResNet-50-FPN-ISO	84.06 ± 0.76	85.78 ± 1.02	83.37 ± 1.00	75.64 ± 0.77	-
	Mask R-DHCNN (**Ours**)	82.47 ± 1.11	85.71 ± 0.79	80.24 ± 2.13	78.07 ± 0.94	5.61
T24	Baseline	92.25 ± 0.83	88.25 ± 0.76	85.18 ± 0.73	93.81 ± 0.56	4.28
	MS R-CNN	91.98 ± 0.07	87.67 ± 0.11	83.41 ± 0.29	93.53 ± 0.31	4.29
	ExtremeNet	81.86 ± 0.88	80.88 ± 0.76	71.54 ± 0.34	79.66 ± 0.50	4.12
	TensorMask	87.53 ± 1.20	83.24 ± 1.09	76.54 ± 1.52	86.33 ± 0.64	2.19
	PolarMask	91.67 ± 0.19	86.08 ± 0.20	83.10 ± 0.44	92.79 ± 0.37	**10.32**
	CenterMask	82.01 ± 1.14	84.80 ± 0.95	73.98 ± 1.08	80.89 ± 0.74	6.42
	ResNet-50-FPN-ISO	93.41 ± 0.66	**92.14 ± 0.61**	83.67 ± 0.71	93.82 ± 0.33	-
	Mask R-DHCNN (**Ours**)	**94.32 ± 0.85**	91.38 ± 0.56	**87.15 ± 0.98**	**94.31 ± 0.54**	6.44

**Table 2 sensors-24-02424-t002:** The influence of DHConv on detection and segmentation results. Bold fonts mark the highest performance.

Datasets	Baseline (Mask R-CNN)	HetConv	DConv	DHConv	PQ	FLOPs (G)	FPS
U373	✓				57.53	55.37	4.81
	✓	✓			57.02	**46.74**	**7.05**
	✓		✓		48.56	55.14	4.34
	✓			✓	**61.24**	46.75	7.00
GoTW1	✓				72.91	73.89	4.00
	✓	✓			72.20	59.46	**6.71**
	✓		✓		61.68	74.06	3.78
	✓			✓	**75.05**	**59.45**	6.70
SIM+-01	✓				68.26	60.23	4.10
	✓	✓			69.11	**50.08**	**5.50**
	✓		✓		53.54	60.38	3.63
	✓			✓	**73.96**	50.13	**5.50**
SIM+-02	✓				46.12	70.01	3.81
	✓	✓			45.10	**58.19**	5.60
	✓		✓		35.68	69.25	3.52
	✓			✓	**49.81**	57.34	4.86
T24	✓				82.87	90.74	4.28
	✓	✓			80.75	**70.85**	**6.45**
	✓		✓		71.58	90.52	3.89
	✓			✓	**84.15**	70.88	6.44

## Data Availability

Data are contained within the article.

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
