# Peer review of "Dilated Heterogeneous Convolution for Cell Detection and Segmentation Based on Mask R-CNN"

_sensors, 2024, doi:10.3390/s24082424_

Round 1
Reviewer 1 Report (New Reviewer)
Comments and Suggestions for Authors
In this paper the authors proposed a model of Mask R-DHCN, which replaces the traditional convolutional kernel in the standard Mask R-CNN with the proposed DHConv module to adapt to shape changes and size differences in cell detection and segmentation tasks, while maintaining competitive FLOPs and FPS. A series of ablation experiments were conducted on various biological cell datasets, and the results showed that the proposed method outperformed some new methods in multiple indicators (AP, Precision, Recall, Dice, and PQ). However, minor modifications must be made before this manuscript is accepted for publication in SENSORS.
Major comments
1. Paper lines 225 to 229: This section mentions the use of P=1/4 and r=2. However, there is no detailed explanation in the article on how to select the parameter P of HetConv and the dilation rate r of Dilated Convolution. Can you supplement the impact of selecting different HetConv parameters P and different Dilated Convolution parameters r on the results?
2. Paper lines 389 to 393: There is one missing item in the ablation experiment, please supplement the results of Baseline (Mask R-CNN) + Dilated Convolution.
Minor comments
3. Paper line 326: This section mentions TN: real cells that are mistakenly identified as non- cell objects. I think that it should be corrected as TN: non-cell objects that are correctly identified as non-cell objects, right?
4. Paper line 328: This section mentions FN: non-cell objects that are correctly identified as non-cell objects. I think that it should be changed to FN: real cells that are mistakenly identified as non-cell objects, right?
5. Paper lines 336 to 337: The calculation formula for Dice is mentioned in lines 336-337 of the paper. I thaink that formula (9) here is incorrect. Please check it.
Author Response
In this paper, the authors proposed a model of Mask R-DHCN, which replaces the traditional convolutional kernel in the standard Mask R-CNN with the proposed DHConv module to adapt to shape changes and size differences in cell detection and segmentation tasks, while maintaining competitive FLOPs and FPS. A series of ablation experiments were conducted on various biological cell datasets, and the results showed that the proposed method outperformed some new methods in multiple indicators (AP, Precision, Recall, Dice, and PQ). However, minor modifications must be made before this manuscript is accepted for publication in SENSORS.
Reviewer’s comments:
Paper lines 225 to 229: This section mentions the use of P=1/4 and r=2. However, there is no detailed explanation in the article on how to select the parameter P of HetConv and the dilation rate r of Dilated Convolution. Can you supplement the impact of selecting different HetConv parameters P and different Dilated Convolution parameters r on the results?
Authors’ response:
We thank the reviewer for her/his great efforts in reviewing our manuscript and providing us with all these insightful suggestions. The idea of HetConv is to replace all the original large-size kernels by 1x1 kernels, except the first one. As the proposed Mask R-DHCNN uses ResNet-50-FPN as its backbone, the composed filters are of four kernels, keeping one kernel of the same size and replacing the rest by 1x1 kernels give P =1/4. In the revised version, we first detail the definition of heterogenous convolution in section 2.2 as “More precisely, the first kernel keeps the size of K × K and the remaining kernels are replaced by small kernels of size 1 × 1 “. In addition, we then state the proposed method uses ResNet-50-FPN as the backbone, the composed filters are of 4 kernels. Keeping one kernel of its original size and replacing the rest with 1x1 kernels give P =1/4 (see blue texts in section 4.1).
Indeed, the dilation was introduced to expand the respective field of the original kernel of size 3x3. However, too large kernel could result in a negative effect on feature extraction. So, we set the dilate rate r=2 was set. In the revised version, we explained the reason for setting r=2 as “In addition, to compensate the adverse effect from small kernels as well as too large kernels, a dilation with rate r = 2 was applied to the 3 × 3 kernels. ”
Reviewer’s comments:
Paper lines 389 to 393: There is one missing item in the ablation experiment, please supplement the results of Baseline (Mask R-CNN) + Dilated Convolution.
Authors’ response:
We thank the dear reviewer for this suggestion. In the revised version, we added the results of Baseline (Mask R-CNN) + Dilated Convolution (DConV) in Table 2 and provided a detailed explanations to the results (see the blue text around Table 2). The dilate convolution does not introduce additional modules, it simply performs the convolution operations with the same number of pixels located at different positions. So, it does not require additional FLOPs. However, the dilation brings adverse effects to the performance. At the same time, the dilated kernel needs to search for pixels at different positions to perform convolution, this operation reduces the speed of model execution.
Reviewer’s comments:
Paper line 326: This section mentions TN: real cells that are mistakenly identified as non-cell objects. I think that it should be corrected as TN: non-cell objects that are correctly identified as non-cell objects, right?
Authors’ response:
The reviewer is absolutely correct. We thank the reviewer for pointing out this mistake. We corrected it in the revised version (see the blue text).
Reviewer’s comments:
Paper line 328: This section mentions FN: non-cell objects that are correctly identified as non-cell objects. I think that it should be changed to FN: real cells that are mistakenly identified as non-cell objects, right?
Authors’ response:
The reviewer is absolutely correct. We made a mistake here. In the revised version, we corrected it.
Reviewer’s comments:
Paper lines 336 to 337: The calculation formula for Dice is mentioned in lines 336-337 of the paper. I think that formula (9) here is incorrect. Please check it.
Authors’ response:
We thank the reviewer for her/his efforts in reviewing our manuscript. Following the suggestion, we checked it carefully and confirmed that formula (9) is correct.
Reviewer 2 Report (New Reviewer)
Comments and Suggestions for Authors
This paper proposes Mask R-DHCNN, a novel instance segmentation method for accurate detection and segmentation of biological cells in microscopy images. The core innovation is a Dilated Heterogeneous Convolution (DHConv) module that combines dilated convolutions to increase receptive fields with heterogeneous kernel sizes to reduce computational costs. By integrating DHConv into the popular Mask R-CNN framework, the authors develop an efficient yet accurate approach for handling variably-shaped and sized cells. Comprehensive experiments demonstrate Mask R-DHCNN's competitive performance against state-of-the-art methods in terms of common metrics like Average Precision and Dice coefficient, while maintaining low computational requirements. To enhance this work, some key questions for the authors are provided:
1. The proposed method is based on the Mask R-CNN framework, which is a two-stage object detection and segmentation model. While the authors claim that the proposed method can reduce computational costs and maintain competitive FLOPs and FPS, it would be beneficial to provide a more detailed analysis of the computational complexity and memory requirements of the proposed method, especially in comparison to other lightweight or efficient instance segmentation models.
2. The authors mention that the proposed method can handle cells with variable shapes and large size differences. However, the experimental results are mainly focused on overall performance metrics such as AP, Precision, Recall, Dice, and PQ. It would be helpful to provide additional qualitative and quantitative analyses to demonstrate the method's robustness to variations in cell shape and size, particularly for challenging cases where cells exhibit highly irregular morphologies or significant size differences within the same image.
3. The authors compare the proposed method with several state-of-the-art instance segmentation methods, such as Mask R-CNN, MS R-CNN, ExtremeNet, TensorMask, PolarMask, and ResNet-50-FPN-ISO. While these comparisons are valuable, it would be beneficial to include additional comparisons with other recent and relevant methods specifically designed for cell detection and segmentation tasks, if available.
4. The authors mention that the proposed method can be adapted to remote devices with limited resources. However, the paper does not provide any specific details or experiments related to the deployment or performance of the method on such devices. It would be valuable to include an analysis or empirical evaluation of the proposed method's performance on resource-constrained devices, such as embedded systems or mobile devices, to support this claim.
5. The authors mention that the proposed method might struggle to capture fine-grained local information due to the dilated convolution and heterogeneous kernel structure. While the authors suggest incorporating clustering methods to handle cells with irregular morphologies, it would be helpful to provide more details on potential strategies or modifications to the proposed method to address this limitation.
Author Response
This paper proposes Mask R-DHCNN, a novel instance segmentation method for accurate detection and segmentation of biological cells in microscopy images. The core innovation is a Dilated Heterogeneous Convolution (DHConv) module that combines dilated convolutions to increase receptive fields with heterogeneous kernel sizes to reduce computational costs. By integrating DHConv into the popular Mask R-CNN framework, the authors develop an efficient yet accurate approach for handling variably-shaped and sized cells. Comprehensive experiments demonstrate Mask R-DHCNN's competitive performance against state-of-the-art methods in terms of common metrics like Average Precision and Dice coefficient, while maintaining low computational requirements. To enhance this work, some key questions for the authors are provided:
Authors’ response:
We thank the reviewer for the efforts in reviewing our manuscript and providing us with very useful and insightful suggestions/critiques, which have all been incorporated in the revised version (see the blue texts). In the following, we provide a point-by-point reply to all these comments.
Reviewer’s comment:
The proposed method is based on the Mask R-CNN framework, which is a two-stage object detection and segmentation model. While the authors claim that the proposed method can reduce computational costs and maintain competitive FLOPs and FPS, it would be beneficial to provide a more detailed analysis of the computational complexity and memory requirements of the proposed method, especially in comparison to other lightweight or efficient instance segmentation models.
Authors’ response:
We thank the reviewer for her/his efforts in reviewing our manuscript. Indeed, theoretical estimation on the reduction of computational cost (Eq.(6)) was given without considering the computations of other modules in the Mask R-CNN structure. So, the actual reduction in FLOPs obtained from the introduction of the HDConv module was less than the theoretical expectation. This implies that when the backbone structure gets more complex, the benefits would be less obvious. In the revised version, we provided a discussion on this point at the end of Section 4 by stating “ Worth mention that the reduction in computational costs (FLOPs) obtained from the introduction of HDConv module was less than the theoretical expectation. This inconsistency resulted from the ignorance of other components in the backbone Mask R-CNN structure when performing the analysis of Eq.(6). Consequently, when the backbone gets larger, the benefits of the HDConv could be less obvious.”
Reviewer’s comment:
The authors mention that the proposed method can handle cells with variable shapes and large size differences. However, the experimental results are mainly focused on overall performance metrics such as AP, Precision, Recall, Dice, and PQ. It would be helpful to provide additional qualitative and quantitative analyses to demonstrate the method's robustness to variations in cell shape and size, particularly for challenging cases where cells exhibit highly irregular morphologies or significant size differences within the same image.
Authors’ response:
We thank the reviewer for this suggestion. In the revised manuscript, we added a figure (Fig 6) to demonstrate the effectiveness and robustness of the proposed method, where the cells in the microscopy image are of different shapes and sizes. The proposed model can segment all these cells out precisely, with only a tiny difference between the prediction and ground truth.
Reviewer’s comment:
The authors compare the proposed method with several state-of-the-art instance segmentation methods, such as Mask R-CNN, MS R-CNN, ExtremeNet, TensorMask, PolarMask, and ResNet-50-FPN-ISO. While these comparisons are valuable, it would be beneficial to include additional comparisons with other recent and relevant methods specifically designed for cell detection and segmentation tasks, if available.
Authors’ response:
Following the reviewer’s suggestion, we added another state-of-the-art method, the CenterMask proposed in 2020, for comparison in Table 1. The proposed model also shows priority over it.
Reviewer’s comment:
The authors mention that the proposed method can be adapted to remote devices with limited resources. However, the paper does not provide any specific details or experiments related to the deployment or performance of the method on such devices. It would be valuable to include an analysis or empirical evaluation of the proposed method's performance on resource-constrained devices, such as embedded systems or mobile devices, to support this claim.
Authors’ response:
We are very sorry for these misleading words. The purpose of the work was to present a way of designing deep neural network-based method for cell detection and segmentation that requires less computational complexity and memory so that its implementation on resource-constrained devices can be achieved. Mask R-CNN is a frequently used and successful solution for object detection and segmentation. A lot of variations have been proposed. We aimed to demonstrate the introduction of HDConv is a plausible solution for reducing computational complexity and memory requirements. In the revised version, we clearly state the purpose of the work in lines 40-44 “ The high computational complexity and memory requirement limits its adaptation in remote devices such as some microscope imaging systems. Consequently, reducing model complexity while keeping its high performance is always plausible. ”
Reviewer’s comment:
The authors mention that the proposed method might struggle to capture fine-grained local information due to the dilated convolution and heterogeneous kernel structure. While the authors suggest incorporating clustering methods to handle cells with irregular morphologies, it would be helpful to provide more details on potential strategies or modifications to the proposed method to address this limitation.
Authors’ response:
We thank the dear reviewer for this very insightful suggestion. In the revised version, we first state that Mask-RCC relies on the introduction of predefined anchor boxes to reduce computation costs. Then, we state that the irregularity of cell morphology makes the predefinition of anchor boxes difficult. The suggested solution is given in the main text as “Clustering methods that generate anchor boxes with different aspect ratios through adaptive learning would be incorporated in the network to handle those cells of irregular shapes [38]. ”
Reviewer 3 Report (New Reviewer)
Comments and Suggestions for Authors
In this paper a novel modofocation of Mask R-CNN model is proposed, which uses a Dilation Heteriogeneous Convolution which integrates the strengths of the heterogeneous kernel structure and dilated convolution. It allows to enlarge the receptive fiels of the convolution without making computations more complex. The heterogeneous kernel structure allows to adapt to different shapes and sizes of object segmentaed. The paper is well written with a lot of numerical experiments that prove the validity of the proposed approach.
The only questions I have, are about stramge font colors, which does not mean anything.
In the examples in Figure 3 and Figure 5 it seems, that all cells are of almost equal sizes (on single image), so the benefits of proposed approach could be not so distinct.
In the table 1 the advantages of proposed method are not so clear, it wins only in some metrics, and in others loose to competitors. So, the more clear enumeration of advantages is desired.
It is unclear to which specific repository leads the Github link that is provided at the end of the paper.
Author Response
In this paper a novel modofocation of Mask R-CNN model is proposed, which uses a Dilation Heteriogeneous Convolution which integrates the strengths of the heterogeneous kernel structure and dilated convolution. It allows to enlarge the receptive fiels of the convolution without making computations more complex. The heterogeneous kernel structure allows to adapt to different shapes and sizes of object segmentaed. The paper is well written with a lot of numerical experiments that prove the validity of the proposed approach.
Authors’ response:
We thank the reviewer for the efforts in reviewing our manuscript and providing us the beneficial and insightful suggestions/critiques, which have all been incorporated in the revised version (see the blue texts). In the following, we provide a point-by-point reply to all these comments.
Reviewer’s comment:
The only questions I have, are about strange font colors, which does not mean anything.
Authors’ response:
We are sorry for not properly formatting the manuscript. In the revised version, we used normal black fonts for the original texts and used blue to indicate the modifications we made in respect to the original one.
Reviewer’s comment:
In the examples in Figure 3 and Figure 5 it seems, that all cells are of almost equal sizes (on single image), so the benefits of proposed approach could be not so distinct.
Authors’ response:
We thank the reviewer for her/his great efforts in reviewing our manuscript. In the revised version, we added another figure (Figure 6) to demonstrate the effectiveness of the proposed method, where cells of different sizes and shapes are detected and segmented.
Reviewer’s comment:
In the table 1 the advantages of proposed method are not so clear, it wins only in some metrics, and in others loose to competitors. So, the more clear enumeration of advantages is desired.
Authors’ response:
We thank the reviewer for this comment and suggestion. The aim of the manuscript was to propose a solution that would keep the advantages of Mask R-CNN in detecting and segmenting cells while being less demanding in computation resources and memory. To explicitly state the purpose of Table 1, we stated that “the proposed method gives quite competitive performance in almost all metrics, with respective to some of the recent state-of-art methods.” The ablation experiments shown below demonstrate that the proposed method achieves these performances without additional computation and memory requirements.
Reviewer’s comment:
It is unclear to which specific repository leads the Github link that is provided at the end of the paper.
Authors’ response:
We apologize for this unclear message. In the revised version, we provided a direct link to the repository of the proposed method, https://github.com/HuHaigen/Mask-R-DHCNN.
Round 2
Reviewer 1 Report (New Reviewer)
Comments and Suggestions for Authors
The authors have done well in revising their manuscript. My recommendation therefore is acceptance in present form.
This manuscript is a resubmission of an earlier submission. The following is a list of the peer review reports and author responses from that submission.
Round 1
Reviewer 1 Report
Comments and Suggestions for Authors
The manuscript proposes a novel approach, Mask R-DHCNN, for addressing the challenges in cell detection and segmentation, particularly focusing on images with variable shapes, large size differences, uneven grayscale, and dense distribution of biological cells. The proposed method introduces the DHConv module, integrating heterogeneous kernel structure and dilated convolution, to enhance adaptability to diverse cell characteristics. Extensive experiments on various biological cell datasets validate the effectiveness of the proposed method, showcasing superior performance compared to state-of-the-art techniques across multiple evaluation metrics. However, several shortcomings should be addressed before the manuscript can be accepted for publication.
1. There is insufficient description of other works related to cell detection and segmentation in the literature review, and further supplementation is recommended.
2. The abbreviations "FPS" and "FLOPs" should be explained the first time they are introduced in the manuscript, rather than waiting until section 4.4.
3. The dataset names displayed in Figure 3 appear to have formatting issues. It is recommended to either use abbreviations or reformat them for clarity.
4. It is suggested to discuss the limitations of the proposed method or potential improvements for future work.
Author Response
Reviewer 1
Reviewer's comment:
The manuscript proposes a novel approach, Mask R-DHCNN, for addressing the challenges in cell detection and segmentation, particularly focusing on images with variable shapes, large size differences, uneven grayscale, and dense distribution of biological cells. The proposed method introduces the DHConv module, integrating heterogeneous kernel structure and dilated convolution, to enhance adaptability to diverse cell characteristics. Extensive experiments on various biological cell datasets validate the effectiveness of the proposed method, showcasing superior performance compared to state-of-the-art techniques across multiple evaluation metrics. However, several shortcomings should be addressed before the manuscript can be accepted for publication.
Authors' response:
We thank the reviewer for her/his great efforts in reviewing our manuscript. We also appreciate the reviewer for her/his comments/suggestions, which have been implemented in the revised version. In the following, we provide a point-by-point reply to these suggestions.
Reviewer's comment:
- There is insufficient description of other works related to cell detection and segmentation in the literature review, and further supplementation is recommended.
Authors' response:
We appreciate your valuable suggestions and have made improvements accordingly. In the revised version, we have added a description of traditional cell detection and segmentation techniques. Specifically, we added the following statement to the introduction on page 1, lines 23-26:
"Traditional methods for cell detection and segmentation, such as thresholding [1,2], edge detection [3], and watershed transform [4-6], have been extensively used in biomedical image analysis. These methods rely on handcrafted features and often require manual tuning of parameters, making them less robust and scalable."
Reviewer's comment:
- The abbreviations "FPS" and "FLOPs" should be explained the first time they are introduced in the manuscript, rather than waiting until section 4.4.
Authors' response:
We thank the reviewer for raising this question. Following your advice, we have made the corresponding revisions to the revised version, relocating the explanations of FPS and FLOPs to the first occurrence in the main text of the article ensuring that readers can grasp the meanings of these terms at the earliest stage possible.
Reviewer's comment:
- The dataset names displayed in Figure 3 appear to have formatting issues. It is recommended to either use abbreviations or reformat them for clarity.
Authors' response:
We apologize for any confusion caused by the unclear presentation. In the revised version, we have abbreviated the names of the images in Figure 3 for better clarity and visualization.
Reviewer's comment:
- It is suggested to discuss the limitations of the proposed method or potential improvements for future work.
Authors' response:
We appreciate the suggestion provided by the reviewer. In the revised version, we have included a section in the "Conclusion and Discussion" where we discuss potential improvements for future work. As “However, it's important to note that the images we are currently using are extracted from standard databases, which are typically collected under favorable conditions. In real-world scenarios, images may present various challenges, which is something we aim to address in future work. Additionally, as Mask R-DHCNN may struggle to capture fine-grained local information, regular-shaped cells are more easily identified and segmented compared to irregular ones. Therefore, we plan to explore incorporating clustering methods to generate anchor boxes with different aspect ratios to further improve the performance in handling irregular cell shapes.”

Reviewer 2 Report
Comments and Suggestions for Authors
Please write a section on how the cells were obtained, how each cell line was cultured, the nature of cell line i.e adherent, non adherent, the imaging conditions, the time point of imaging post harvesting, the type of microscope used, objective lens, how were the slides prepared for imaging etc.
Please add scale bars to all images
Please add a bigger panel of representative cell types and show the detection in every panel. Figure 4 should be a bigger panel showing images and detection on all the different kind of imaging conditions you captured.
Please discuss more on if a certain imaging condition was more favorable as compared to that of the other
Please discuss more on if a certain kind of cell line was more easier to detect as compared to others. Please add some correlations on the biology of cells and the efficiency of detection.
Please add more information of average size of each group of cell. Were you able to differentiate between the nucleus of the cell and the rest of cell body? Please provide a discussion on how this can be achieved.
Please add additional discussion on different features noticed in the cells. It is important to note that several studies have worked on cell detection, however, there is still lack of understanding about cell features in these different cell types.
Please explain what do you mean by two image sequences? Does it mean two images of the same spot? Or two different filters?
How do you define high density, low contrast, please elaborate and categorize the limits appropriately. Please provide numeric data and relevant references.
Would this detection technique still work if cells were fixed using 4% formaldehyde or other fixation agents?
Please write a section on challenges faced.
How would your detection technique differentiate between a cell and a debris?
Please insert a table showing manual counts v/s detected counts
Please add a discussion section on how this would translate in blood of real cancer patients? Would you be able to distinguish between RBCs, WBCs and CTCs?
Please perform more experiments by spiking these cancer cells in blood samples, and see if you detection algorithm still stands. There are several methods detecting pure cell populations, however the real challenge stands in identifying cancer cells in a mixed cell population thus mimicking real life scenario.
Author Response
Reviewer's comment:
Please write a section on how the cells were obtained, how each cell line was cultured, the nature of cell line i.e adherent, non-adherent, the imaging conditions, the time point of imaging post harvesting, the type of microscope used, objective lens, how were the slides prepared for imaging etc.
Authors' response:
We appreciate the suggestions provided by the reviewer. In the revised version, we have provided further detailed explanations regarding the sources of the cells in the dataset within section 4.2 "Datasets". For example, we have enhanced descriptions such as: “PhC-C2DH-U373 (U373) comprises images of glioblastoma-astrocytoma cells obtained through phase contrast microscopy. These cells are cultured adherently and imaged under controlled conditions to provide data for cell tracking and analysis tasks.”” Fluo-N2DH-GOWT1 (GOWT1) was obtained from GFP transfected GOWT1 mouse embryonic stem cells on a flat substrate using a Leica TCS SP5 laser scanning confocal microscope (courtesy of Dr E. Bártová, Academy of Sciences of the Czech Republic, Brno, Czech Republic). This dataset exhibits characteristics of low contrast and high density under the microscope due to factors such as heterogeneous staining, prominent nucleoli, mitoses, cells entering and leaving the field of view, and frequent cell collisions.”” T24 is a real bladder cancer cell dataset from the Shanghai Institute of Cell Biology, Chinese Academy of Sciences (Shanghai, China). The cells were cultured in RPMI-1640 medium (HyClone, Logan, UT, USA) supplemented with 10% heat-inactivated FBS (JRH Biosciences, Lenexa, KS, USA),100 U/mL penicillin, and 100 mg/L streptomycin. Cultures were maintained in a humidified atmosphere of 5% CO2 at 37℃.”
Reviewer's comment:
Please add scale bars to all images.
Authors' response:
We appreciate the efforts the reviewer put in reviewing our manuscript. The purpose of the current manuscript is to build artificial intelligence-based tool to help biologists in detecting and segmenting cells from the microscopy images. These microscopy images used in this study are from a publicly available database(the ISBI2015 Cell Tracking Challenge), specifically designed for computer scientists for developing tools. So, cells in these images were marked by biologists for providing the ground truth to validate the effectiveness of the many automatic methods. For this reason, the original dataset does not have the information on the scale of images. We agree that adding scale bars would be helpful to biologists. We discuss the effect of the lack scale information of images might cause in the last paragraph of the manuscript as” A common case is difference caused by different scales, as some detailed morphological structures are not visible under small scales. Unfortunately, there is no scale information of the images provided in the database. No suggestions would be given to biologists on how to collect microscopy images. ”
Reviewer's comment:
Please add a bigger panel of representative cell types and show the detection in every panel. Figure 4 should be a bigger panel showing images and detection on all the different kind of imaging conditions you captured.
Authors' response:
We thank the dear reviewer for her/his suggestions. Indeed, the original figure was very blurred and not clear enough. Following her/his suggestion, we reproduced the figure and enlarged it in the manuscript.
Reviewer's comment:
Please discuss more on if a certain imaging condition was more favorable as compared to that of the other
Authors' response:
We appreciate the suggestion provided by the reviewer. In the revised version, we have included a section in the "Conclusion and Discussion" where we added a discuss on the scaling of microscopy images as well as the contamination of dusts. In the last paragraph of the manuscript, we first state the scaling information is not provided in the database, while different morphological details appear under different scales. So the proposed method would work well on images of the same scale, and fine-tune might be needed to adapt to different scales. We then discuss the problem of dust contamination.
Reviewer's comment:
Please discuss more on if a certain kind of cell line was more easier to detect as compared to others.
Authors' response:
We appreciate the suggestion provided by the reviewer. In the revised version, we have included a section in the "Conclusion and Discussion" where we discuss potential improvements for future work. As “In addition, as Mask R-DHCNN may struggle to capture fine-grained local information, the proposed method might be less effective in detecting and segmenting cells of irregular morphological structure comparing to those of common regular shapes. Clustering methods that generate anchor boxes with different aspect ratios would be incorporated in the network to handle those cells of irregular shapes.”
Reviewer's comment:
Please add some correlations on the biology of cells and the efficiency of detection.
Authors' response:
We thank the reviewer for this very insightful suggestion. Following her/his suggestion, we added a sentence in the introduction part as “ For example, quick detection of leucocyte is crucial to the early diagnosis of infections” to better illustrating the motivation of the current work.
Reviewer's comment:
Please add more information of average size of each group of cell. Were you able to differentiate between the nucleus of the cell and the rest of cell body? Please provide a discussion on how this can be achieved.
Authors' response:
We appreciate the dear reviewer’s great efforts in reviewing our manuscript. Due to the lack of scaling information of the microscopy images in the dataset, we could not have the information on the size of cells. We provided a short discussion on the scaling information of the images in the last paragraph of the manuscript.
As reflected in the title of our manuscript, the purpose of the current work is to detect and segment cells from its environment using deep neuron networks approach. The key idea is on how to design such a network that is capable of detecting and segmenting cells. Differentiating nucleus from the rest of the cell body is out of the scope of the current.
Reviewer's comment:
Please add additional discussion on different features noticed in the cells. It is important to note that several studies have worked on cell detection, however, there is still lack of understanding about cell features in these different cell types.
Authors' response:
We thank the reviewer for her/his great efforts in reviewing our manuscript. As manifested in the title, our work mainly focuses on the design and validation of a novel deep neuron network in detecting and segmenting cells from microscopy images. The key advantage of using deep neuron network in this kind work is its automatic extraction of sematic features. However, the extracted features are embedded in the network, no explicit features can be shown. In the revised version, we stated shortcoming of using the handcrafted features in detection and segmentation tasks in the introduction part( line 22-25).
Reviewer's comment:
Please explain what do you mean by two image sequences? Does it mean two images of the same spot? Or two different filters?
Authors' response:
We apologize for this unclear expression. As the dataset was created for cell tracking competition, the cultured cells were video recorded, and each frame of the video can be considered as a microscopy image of the cells. However, the subtle difference between two video frames (what we meant by two image consequences) makes inappropriate use of each frame as an effective data sample. In the revised version, we rephased the description on how we prepared this dataset for cell detection and segmentation task from the original U373.
While the “two image sequences” in the description of dataset Fluo-N2DH-SIM+ means the image of simulated cells and the counterpart stained with fluorescent chromosomes. In the revised version, we rephased it to avoid misunderstanding.
Reviewer's comment:
How do you define high density, low contrast, please elaborate and categorize the limits appropriately. Please provide numeric data and relevant references.
Authors' response:
We thank the reviewer for her/his great efforts put in our manuscript. These datasets used in the current study are standard ones for the cell tracking challenge. Different datasets mark different levels of difficulties, which were defined according to the reference of Ulman et al. (Nature methods 2017, 14, 1141–1152). In the revised version, we rephased the sentence to explicitly state “the characteristics of these datasets according to the criteria given in [38]”.
Reviewer's comment:
Would this detection technique still work if cells were fixed using 4% formaldehyde or other fixation agents?
Authors' response:
The reviewer has raised a very important issue of the deep neuron network-based solutions to object detection and segmentation, the robustness. In the current study, we validated the proposed method in different datasets of quite different cell types and morphology, to demonstrate its robustness. As we stated in the last section (Conclusion and Discussion), its direction adaption of other cells or microscopy images prepared under different conditions might need fine tuning or retraining.
Reviewer's comment:
Please write a section on challenges faced.
Authors' response:
We thank the reviewer for her/his great efforts put on our manuscript. Following her/his suggestion, we added paragraphs to discuss the challenges faced in the last part of the revised manuscript. These challenges include adaptation to different cells and environment conditions, interpretability, and lack of local information of cell boundary.
Reviewer's comment:
How would your detection technique differentiate between a cell and a debris?
Authors' response:
We thank the reviewer for her/his great efforts put on our manuscript. The deep neuron network is black box. When trained with enough well labeled samples, the network would effectively differentiate a cell and a debris. However, lacking interpretability is indeed one of the main shortcomings of the deep neuron networks. The results shown in our manuscript demonstrate that the proposed network could detect cells from different environmental conditions which might be of debris. However, the dataset has no information on the appearance of debris. Further studies are required. In the revised version, we added a short discussion on the effect of debris (dusts) in the last part of the manuscript.
Reviewer's comment:
Please insert a table showing manual counts v/s detected counts
Authors' response:
We thank the reviewer for her/his great efforts put on our manuscript. The current study focuses on the design of deep neuron network that is capable of detecting and segmenting cells from the microscopy images. To validate the effectiveness of the network, the detections and segmentation results given by the network were compared directly with the ground truths, which were the detecting and segmentation results given by manual operations. Recalling the definition of metrics in sections 4.4, the manual count was exactly the sum of true positive and false negative. So, the Recall metric defined in equation (9) gives numerical and more direct or standard comparison between the manual counts and model detection. This quantity was in table 2, with values higher than 0.8 demonstrating the effectiveness of the proposed method.
Reviewer's comment:
Please add a discussion section on how this would translate in blood of real cancer patients? Would you be able to distinguish between RBCs, WBCs and CTCs?
Authors' response:
We thank the reviewer for her/his great efforts put on our manuscript. As manifested in the title, our work was intent on providing a deep neural network-based tools for biologists in detecting and segmenting cells from the microscopy images they collect. As networks trained with specific data samples are typically suitable for specific tasks. Adaptation to other tasks typically requires parameter fine tuning or even retraining. In the current study, we only used data samples in some public available cell microscope images, no information on cell types and other biological/clinical information was provided. As a results, the model could not be directly adapted in other tasks, such as distinguishing different cell types. In the revised version, we discussed this point in the last part of the manuscript as “Since the 380 images in the database were typically collected under some specific conditions, the network 381 trained with these images might need fine tune to adapt to real-world scenarios.”
Reviewer's comment:
Please perform more experiments by spiking these cancer cells in blood samples, and see if you detection algorithm still stands. There are several methods detecting pure cell populations, however the real challenge stands in identifying cancer cells in a mixed cell population thus mimicking real life scenario.
Authors' response:
We really appreciate the dear reviewer’s hard working. The purpose of the manuscript is to design/provide a new tool for biologists in cell detection and segmentation tasks. As computer scientists, we are not capable of doing biological experiments by ourselves. We hope that our work would help or facilitate more advanced biological exploration. To this end, we showed the results obtained on public available datasets to inspire adaptation and improvements by biologists and technicians. To better serve this purpose, we made our codes available on Github to encourage real applications. We explicitly stated this in the revised manuscript.
Reviewer 3 Report
Comments and Suggestions for Authors
This manuscript introduces an innovative convolution kernel structure, DHConv (Dilation Heterogeneous Convolution), aimed at improving cell detection and segmentation in microscopy images. The proposed method addresses challenges associated with the standard Mask R-CNN, including the large number of learning parameters and computational demands. By integrating the advantages of heterogeneous kernel structures and dilated convolutions, DHConv enhances adaptability to variable shapes and sizes of biological cells. The effectiveness of this approach is demonstrated through comparative and ablation studies across various biological cell datasets, showing superior performance in some instances over existing methods.
**Major Concerns:**
1. **Validation on Diverse Datasets:** The manuscript primarily focuses on the performance of DHConv on certain datasets. Including more varied datasets, especially those with more challenging and diverse imaging conditions, would strengthen the claims of versatility and effectiveness.
2. **Comparison with State-of-the-Art Methods:** While the manuscript mentions superior performance in some cases, a more exhaustive comparison with current state-of-the-art methods, including quantitative metrics and qualitative visual comparisons, would provide a clearer understanding of the method's advantages and limitations.
3. **Parameter Optimization and Computational Efficiency:** The manuscript briefly touches upon the computational advantages of DHConv. However, a detailed analysis comparing the computational efficiency and parameter optimization with existing methods would be beneficial for readers, especially considering the application in resource-limited settings.
4. **Real-World Application and Scalability:** The manuscript would benefit from a discussion on the scalability of DHConv and its application in real-world microscopy imaging scenarios. Insights into the deployment challenges and potential integration with existing microscopy imaging workflows would be valuable.
5. **Methodological Details and Reproducibility:** Some sections of the manuscript could benefit from a more detailed explanation of the methodology, including the architecture specifics of DHConv, to enhance reproducibility. Additionally, making the code available publicly could foster further research and application of the proposed method.
**Minor Concerns:**
1. **Literature Review:** The literature review could be expanded to include recent advancements in the field of cell detection and segmentation, providing a broader context for the study's contributions.
2. **Graphical and Visual Representations:** Including more visual comparisons between DHConv and other methods could aid in understanding the qualitative improvements. Enhanced visual aids, such as segmentation maps and detection examples, would be beneficial.
3. **Statistical Analysis:** The manuscript would benefit from a more thorough statistical analysis to support the comparative performance claims, including confidence intervals or p-values where applicable.
Author Response
Reviewer's comment:
This manuscript introduces an innovative convolution kernel structure, DHConv (Dilation Heterogeneous Convolution), aimed at improving cell detection and segmentation in microscopy images. The proposed method addresses challenges associated with the standard Mask R-CNN, including the large number of learning parameters and computational demands. By integrating the advantages of heterogeneous kernel structures and dilated convolutions, DHConv enhances adaptability to variable shapes and sizes of biological cells. The effectiveness of this approach is demonstrated through comparative and ablation studies across various biological cell datasets, showing superior performance in some instances over existing methods.
Authors' response:
We thank the reviewer’s hard work. We also appreciate the reviewer’s suggestions/comments/critiques, which helped a lot for us to improve the quality of the manuscript. In the following, we provide point-by-point explanations on how we implemented these suggestions in the revised manuscript.
Major Concerns:
Reviewer's comment:
- **Validation on Diverse Datasets:** The manuscript primarily focuses on the performance of DHConv on certain datasets. Including more varied datasets, especially those with more challenging and diverse imaging conditions, would strengthen the claims of versatility and effectiveness.
Authors' response:
We thank the reviewer for this suggestion. In this work, we proposed a novel deep neural network called Mask R-DHConv to accomplish cell detection and segmentation tasks. We test its performance on the most frequently used and standard 5 microscopy cell image databases. The obtained performance on these datasets validated the effectiveness of the method. Indeed, including more varied datasets would further confirm the versatility and effectiveness. Due to the limited availability of those datasets, we were not able to find additional datasets that were suitable for further verifications. However, we included additional 3 different deep neural network-based solutions to the same task on the same datasets, namely CenterMask, YoLACT, and PolarMask to illustrate the effectiveness of the proposed method.
Reviewer's comment:
- **Comparison with State-of-the-Art Methods:** While the manuscript mentions superior performance in some cases, a more exhaustive comparison with current state-of-the-art methods, including quantitative metrics and qualitative visual comparisons, would provide a clearer understanding of the method's advantages and limitations.
Authors' response:
We thank the reviewer for this suggestion. Following her/his suggestion, we added 2 additional deep neural network-based solutions to the same task on the same datasets, namely PolarMask and ResNet-50-FPN-ISO to illustrate the effectiveness of the proposed method. Detailed comparisons to these state-of-the-art methods are given in Table 2.
Reviewer's comment:
- **Parameter Optimization and Computational Efficiency:** The manuscript briefly touches upon the computational advantages of DHConv. However, a detailed analysis comparing the computational efficiency and parameter optimization with existing methods would be beneficial for readers, especially considering the application in resource-limited settings.
Authors' response:
We thank the reviewer’s great efforts put into our manuscript. Her/his suggestions are very helpful to the manuscript. Following this suggestion, in the experimental settings section (page 6, section 4.1) we detailed the parameters of the network and the potential computational efficiency the proposed method would bring (see the blue texts in lines 200-204).
Reviewer's comment:
- **Real-World Application and Scalability:** The manuscript would benefit from a discussion on the scalability of DHConv and its application in real-world microscopy imaging scenarios. Insights into the deployment challenges and potential integration with existing microscopy imaging workflows would be valuable.
Authors' response:
We thank the reviewer for her/his great efforts in our manuscript. Following the suggestions, we added sentences to discuss the possibility of applying the proposed method in real-world microscopy images. These discussions are marked in blue in the last second paragraph of the last section. Need to say, our work only serves as a motivation for new innovative technologies. The current model could not be integrated into a microscopy setting. In the revised version, we explicitly stated this point in the last sentence of our manuscript.
Reviewer's comment:
- **Methodological Details and Reproducibility:** Some sections of the manuscript could benefit from a more detailed explanation of the methodology, including the architecture specifics of DHConv, to enhance reproducibility. Additionally, making the code available publicly could foster further research and application of the proposed method.
Authors' response:
We thank the reviewer for giving us these useful /helpful suggestions. Following the suggestions, we detailed the DHConv setting we used in the current study in Section 4.1. To enhance reproducibility, we also made our codes available on Github, with a link provided in the last part of the manuscript.
**Minor Concerns:**
Reviewer's comment:
- **Literature Review:** The literature review could be expanded to include recent advancements in the field of cell detection and segmentation, providing a broader context for the study's contributions.
Authors' response:
We appreciate your valuable suggestions and have made improvements accordingly. In the revised version, we have added a description of traditional cell detection and segmentation techniques. Specifically, we added the following statement to the introduction on page 1, lines 23-26:
"Traditional methods for cell detection and segmentation, such as thresholding [1,2], edge detection [3], and watershed transform [4-6], have been extensively used in biomedical image analysis. These methods rely on handcrafted features and often require manual tuning of parameters, making them less robust and scalable."
Reviewer's comment:
- **Graphical and Visual Representations:** Including more visual comparisons between DHConv and other methods could aid in understanding the qualitative improvements. Enhanced visual aids, such as segmentation maps and detection examples, would be beneficial.
Authors' response:
We thank the dear reviewer for her/his hard work and suggestions. Following those, we added a new figure (Fig. 5) to present a visual comparison of how the proposed model detects and segments cells in microscopy images.
Reviewer's comment:
- **Statistical Analysis:** The manuscript would benefit from a more thorough statistical analysis to support the comparative performance claims, including confidence intervals or p-values where applicable.
Authors' response:
We thank the reviewer’s suggestion, following which we have added the standard deviations to each of the metrics shown in Table 2 in the revised version.
Round 2
Reviewer 2 Report
Comments and Suggestions for Authors
Thank you for addressing my comments